# Single-Step Self-Assembly and Physical Crosslinking of PEGylated Chitosan Nanoparticles by Tannic Acid

**DOI:** 10.3390/polym11050749

**Published:** 2019-04-27

**Authors:** Raven A. Smith, Rebecca C. Walker, Shani L. Levit, Christina Tang

**Affiliations:** Chemical and Life Science Engineering Department, Virginia Commonwealth University, Richmond, VA 23284-3028, USA; smithra7@mymail.vcu.edu (R.A.S.); walkerrc4@mymail.vcu.edu (R.C.W.); levitsl@mymail.vcu.edu (S.L.L.)

**Keywords:** Flash NanoPrecipitation, tannic acid, chitosan, crosslinking, nanoparticles

## Abstract

Chitosan-based nanoparticles are promising materials for potential biomedical applications. We used Flash NanoPrecipitation as a rapid, scalable, single-step method to achieve self-assembly of crosslinked chitosan nanoparticles. Self-assembly was driven by electrostatic interactions, hydrogen bonding, and hydrophobic interactions; tannic acid served to precipitate chitosan to seed nanoparticle formation and crosslink the chitosan to stabilize the resulting particles. The size of the nanoparticles can be tuned by varying formulation parameters including the total solids concentration and block copolymer to core mass ratio. We demonstrated that hydrophobic moieties can be incorporated into the nanoparticle using a lipophilic fluorescent dye as a model system.

## 1. Introduction

Chitosan is a cationic polysaccharide derived from deacetylation of chitin (abundant in crustacean shells). Chitosan has been widely considered for medical and pharmaceutical applications because it is non-toxic, biocompatible, and biodegradable. Due to the amino group content, chitosan is insoluble at neutral and alkaline pHs. Upon protonation of the amino groups, chitosan is soluble in acidic media. This pH dependent solubility in chitosan micro-/nanoparticle systems is promising for controlled release applications [1,2]. Additionally, the resulting cationic particles have antimicrobial [3] and mucoadhesive properties [4]. Recently, cationic particles showed enhanced immune response and are considered especially promising for pulmonary delivery [5] and immunotherapies [6].

Several methods have been developed to produce chitosan nanoparticles, typically involving electrostatic interactions or emulsion-based processes. Electrostatically assembled nanoparticles can be achieved by ionic gelation which utilizes interactions between the chitosan amine groups and a polyanion such as tripolyphosphate. Similarly, chitosan interactions with anionic macromolecules such as DNA can be leveraged to self-assemble polyelectrolyte complexes/coacervates. Chitosan nanoparticles can also be achieved using emulsion-based processing. For example, nanoparticles can be formed from water-in-oil microemulsions in which aqueous chitosan droplets are dispersed (by applying high shear) in an organic solvent containing a covalent crosslinking agent (e.g., glutaraldehyde) and evaporating the organic solvent. These aqueous based methods that minimize the use of solvents are advantageous for incorporating negatively charged biomolecules such as oligonucleotides, proteins, and peptides. The resulting particles can be used for a variety of applications such as mucosal delivery of biomolecules [1,7,8]. However, alternative methods are needed to accommodate hydrophobic moieties of interest, e.g., small molecule drugs [2,4].

For example, incorporation of hydrophobic small molecules into chitosan particle systems has also been achieved by making the chitosan particles using the methods described above, crosslinking the chitosan, and then soaking the pre-made particles in concentrated solutions of the hydrophobic compound [9]. Double emulsion methods have also been considered. Such methods involve forming an oil (drug)-in-water (chitosan) emulsion which is then emulsified with paraffin to create an oil–water–oil emulsion. The resulting emulsion droplets are dried and crosslinked to generate the drug-loaded chitosan particles [2]. Self-assembly of chitosan-derivatives has also been achieved, which requires first producing the derivatives then processing the derivatives into nanoparticles [1,7]. The disadvantage of these approaches is that they inherently require multiple processing steps that require minutes to days which limits them to the laboratory scale. Rapid (i.e., within seconds), single-step, scalable methods to produce chitosan nanoparticles that incorporate hydrophobic compounds would be beneficial to facilitate broad application of such particles [8].

Flash NanoPrecipitation (FNP) is a well-established method for rapid, polymer-directed self-assembly. Traditionally, FNP encapsulation of hydrophobic moieties is achieved by precipitation of the hydrophobic compound of interest in the presence of a micellizing amphiphilic diblock co-polymer. Nanoparticle assembly occurs due to hydrophobic interactions of the precipitating core material and the hydrophobic block of the block copolymer. Micromixing achieved using specialized mixing geometries such as the multi-inlet vortex mixer or confined impinging jet mixers leads to uniform particle size [10,11,12,13,14]. 

While FNP with highly hydrophobic materials is well established [10,11,12,13,14], recent advances in FNP have expanded materials that can be incorporated into nanoparticles during FNP by leveraging other material interactions such as coordination complexation or electrostatic interactions to encapsulate less hydrophobic materials such as tannic acid (TA) or ion pairs of an ionizable compound and a hydrophobic salt, respectively [15,16]. Other charged macromolecules have been incorporated into FNP as stabilizers. For example, rapid mixing and precipitation reactions have been combined to produce lead sulfate nanoparticles stabilized by a chitosan shell [17]. Hydrophobic zein protein nanocarriers stabilized by casein have also been reported [18]. The zein defined the nanoparticle size and the particle was stabilized by casein interactions. Interestingly, the zein nanocarriers were stable in buffer and swelled at pH 2 when ionic complexation was eliminated indicating that hydrophobic interactions maintain particle integrity. Recently, Flash NanoComplexation has been reported for continuous production of DNA nanoparticles stabilized by polyethylenimine [19]. However, the ability to rapidly produce PEGylated chitosan particles via FNP has not been demonstrated. 

These previous studies indicate that Flash NanoPrecipitation leveraging electrostatic interactions may be a promising platform to achieve rapid self-assembly of PEGylated chitosan-based nanoparticles. Therefore, in this work, we use FNP as a rapid (i.e., within seconds), one-step method to self-assemble PEGylated chitosan nanoparticles. The focus of this work is to determine the intermolecular forces that drive nanoparticle self-assembly. The effect of formulation parameters on nanoparticle size is examined. The effect of intermolecular interactions on particle stability as a function of pH are discussed. Incorporating a lipophilic fluorescent dye into the nanoparticles during mixing as a model hydrophobic compound is also explored. 

## 2. Materials and Methods 

### 2.1. Materials

Tannic acid (ACS reagent grade), chitosan (75–85% deacetylated with molecular weight 50,000–190,000 Da), nile red (technical grade), and hydrochloric acid, HCl, (37%, ACS reagent grade) were obtained from Sigma–Aldrich (St. Louis, MO, USA). The ACS reagent grade acetone and tetrahydrofuran (THF) were obtained from Fisher Scientific (Pittsburgh, PA, USA). These chemicals were used as received. Amphiphilic block copolymer PS-b-PEG (1600-b-5000 g/mol) was obtained from Polymer Source (Quebec, Dorval, Canada). Prior to use, PS-b-PEG was dissolved in tetrahydrofuran (THF) (500 mg/mL) and precipitated in ether (~1:20 THF:ether ratio). The PS-*b*-PEG was recovered by centrifuging, decanting, and drying under vacuum at room temperature for 2 days.

### 2.2. Nanoparticle Assembly

Nanoparticle assembly was performed using a confined impinging jet mixer with dilution as previously described [20,21]. Under hand operation with 1 mL syringes, mixing Reynold’s numbers of 1300 were achieved [15,22]. Briefly, to achieve PEGylated chitosan-based nanoparticles, block copolymer and TA were dissolved in a water miscible organic solvent (e.g., acetone) at 20 mg/mL and 10 mg/mL, respectively. The organic solvent stream (0.5 mL) containing molecular dissolved block copolymer and TA was rapidly mixed against an equal volume of an aqueous stream containing 10 mg/mL chitosan dissolved in 1.0 M HCl using a confined impinging jet mixer. The mixed stream was collected in 4 mL of water so that the final dispersion contained 10 vol % organic solvent. Hand operation requires less than a second, and nanoparticle self-assembly occurs in the internal mixing chamber of the confined impinging jet mixer within milliseconds [20]. In some cases, nile red incorporated into the nanoparticles during FNP by dissolving it in the organic solvent stream along with the block copolymer and TA. 

### 2.3. Nanoparticle Characterization 

Nanoparticle size distributions and zeta potential were measured by dynamic light scattering (DLS, Zetasizer Nano-ZS, backscatter detection angle of 173°, Malvern Instruments Ltd., Malvern, UK). Size distributions are reported using the normal resolution mode intensity weighted distribution (average of four measurements). The size reported is the peak 1 mean intensity as previously described [15]. The polydispersity index (PDI) is a measure of the breadth of size distribution, was determined from the instrument software (appropriate for samples with PDI < 0.3) and was used as a measure of particle uniformity. Unless otherwise specified, the solvent for DLS was water. 

Nanoparticle stability was evaluated by monitoring the nanoparticle size distributions at various times after mixing. For measuring nanoparticle size stability as a function of pH, dialysis was performed using regenerated cellulose tubing with a molecular weight cutoff of 6–8 kDa (Spectra/Por, Spectrum Laboratories) against aqueous media at various pHs. Aqueous HCl or NaOH at appropriate concentrations were used to achieve acidic or basic media with specified pHs, respectively. Following dialysis, the size of the nanoparticles at the various pH was measured byDLS.

For fluorescence measurements, 100 microliters of the aqueous nanoparticles dispersions resulting from FNP with fluorescent dye were diluted 900 microliters of water. This dilution minimizes the effect of light scattering from the nanoparticles on fluorescence intensity as has been previously established [23]. Emission spectra (550–800 nm) of the nanoparticle dispersions were measured at an excitation wavelength of 530 nm using a Cary Eclipse (Varian, Agilent, Santa Clara, CA, USA).

## 3. Results and Discussion

To rapidly (i.e., within seconds) achieve PEGylated chitosan nanoparticles, we performed Flash NanoPrecipitation (FNP) with chitosan and a block copolymer stabilizer, polystyrene-*b*-polyethylene glycol (PS-*b*-PEG). Since chitosan is soluble at acidic pH and insoluble at higher pH, we initially dissolved chitosan in HCl (pH 2) and mixed it with PS-*b*-PEG dissolved in acetone using a confined impinging jet mixer. The effluent of the mixer was immediately diluted with water to increase the pH and reduce the solubility of chitosan. However, when chitosan and block copolymer undergo FNP, the chitosan is not sufficiently insoluble to be encapsulated. Rather, the chitosan remains soluble and the block copolymer forms micelles evident by the red trace in Figure 1 compared to PS-*b*-PEG micelles formed by FNP shown in black in Figure 1. Additionally, low affinity between polystyrene and cationic core materials has previously observed to result in formation of empty micelles [15]. Given the cationic charge of chitosan, we sought alternative approaches to performing FNP with chitosan.

Our approach was to leverage tannic acid (TA) and chitosan interactions to facilitate nanoparticle self-assembly. Specifically, tannins such as TA precipitate macromolecules including proteins and some polysaccharides such as chitosan. Therefore, we investigated a complexation approach recently developed for encapsulation of hydrophobic ion pairs [16] or TA-iron coordination complexes [15]. Specifically, we added TA to the formulation to precipitate the chitosan during mixing to initiate nanoparticle self-assembly and stabilize the precipitate with hydrophobic interactions with an amphiphilic block copolymer.

Initially, we examined the time scale of TA and chitosan precipitation under the solvent conditions of interest for Flash NanoPrecipitation (FNP). Tannic acid dissolved in acetone and chitosan dissolved in HCl (pH 1–2) were mixed using a confined impinging jet mixer. White precipitate was observed immediately (Figure 1). This result indicates that the time scale of precipitation is sufficiently rapid to facilitate nanoparticle self-assembly via FNP validating our approach. We examined the effect of the ratio of TA to chitosan by mixing various amounts of the two components and observing the amount of precipitate that formed macroscopically. The largest amount of precipitate formed at a 1:1 ratio of TA to chitosan (by mass) and was, thus, used as the basis for forming nanoparticles. 

Next, FNP was performed by mixing an organic stream of TA and amphiphilic PS-*b*-PEG dissolved in acetone with an aqueous stream of chitosan dissolved in HCl solution (pH 2.0) and immediately diluting with water. After FNP with the presence of the block copolymer, results in a colloidal dispersion (opalescent) instead of white precipitate of TA and chitosan. Photographs of colloidal dispersion achieved with the block copolymer stabilizer compared to the TA-chitosan precipitate that collects at the bottom of the vial is shown in Figure 2. This result indicates that the block copolymer serves to stabilize the nanoparticle.

We characterized the size and surface change of the resulting nanoparticle dispersion by dynamic light scattering (DLS). In the presence of the block copolymer, particles formed and were 920 ± 150 nm in diameter and the particle population was uniform as only a single Gaussian peak was apparent using DLS and the PDI was 0.21 ± 0.02 (green trace, Figure 1). The zeta potential of the resulting nanoparticle dispersion was +16 ± 4 mV, which is lower than typically reported for chitosan particles (~+50 mV) [24,25]. This result also suggests that the nanoparticles were to some degree sterically stabilized by the block copolymer as zeta potentials greater than +35 mV are required for fully electrostatic colloidal stabilization [21]. 

Overall, the ability to produce sterically stabilized chitosan-TA precipitate nanoparticles is consistent with previous reports leveraging ion pairing or coordination complexation to form nanoparticles via FNP [15,16]. Given the chitosan-TA precipitate is sterically stabilized by the block copolymer, nanoparticle assembly initiates when TA precipitates with chitosan and the block copolymer stabilizes the precipitate via hydrophobic interactions. Under rapid mixing, we note that TA also interacts with the block copolymer. Specifically, TA and the PEG-block of the block copolymer complex due to hydrogen bonding and this insoluble complex was also confined to the nanoparticle core. Nanoparticles were formed when the hydrophobic block of the block copolymer adsorbed to the precipitating core materials. The nanoparticle core was sterically stabilized by adsorption of the PS-block of the block copolymer due to hydrophobic interactions. The hydrophobic PS blocks anchored onto the insoluble core made of TA-chitosan precipitate, TA:PEG, and the PS-block. The free (not complexed with TA) PEG chains oriented into the aqueous phase microphase separated from the PS-block and sterically stabilizes the nanoparticle. Overall, PEGylated chitosan nanoparticle assembly was driven by a combination of hydrophobic and hydrogen bonding associations. 

Next, we investigated the effect of formulation parameters on nanoparticle size. The DLS size intensity distributions at various block copolymer to core ratios is shown in Figure 3. As is typically observed in FNP, smaller particles could be obtained by increasing the block copolymer to core ratio (by mass). This trend has generally been attributed to geometric arguments as decreasing the core mass (volume) relative to the BCP (surface area) will result in smaller particles nanoparticles [26]. In the TA-chitosan system, ~920 ± 150 nm particles with a PDI of 0.21 ± 0.2 were initially formed using a mass ratio of 1:2 block copolymer to core (red trace in Figure 3). By increasing the amount of block copolymer so that the mass ratio was 1:1 the particle size was ~120 ± 11 nm with a PDI 0.084 ± 0.017 (blue trace in Figure 3). In both of these cases, the particle population was uniform as indicated by the single Gaussian peak evident in the size intensity distributions measured by DLS. 

Further increasing the block copolymer concentration to 5:4 block copolymer to core resulted in two size populations, i.e. multiple peaks in the size intensity distribution (green trace in Figure 3). We attribute the peak at lower size (~30 nm) to the formation of empty micelles. The formation of empty micelles occurs at increased block copolymer concentration due to the limited affinity between the polystyrene block of the block copolymer and the TA-chitosan precipitate. When there is low affinity between the hydrophobic block and the core material, the block copolymer rapidly forms micelles and on a longer time scale stabilizes the precipitating TA–chitosan complex [15] resulting in the bimodal size distribution comprised of the ~275 nm PEGylated chitosan particles and empty micelles (~30 nm). We note that the range of block copolymer to core ratio that form uniform particles is relatively narrow compared to the range used with hydrophobic core materials [26], which is consistent with particle formation involving coordination complexation [15].

We also investigated the effect of total solids concentration on nanoparticle size. The size of the nanoparticles could be tuned between ~100 and 250 nm by varying the total solids concentration from 0.5 mg/mL to 6 mg/mL in the final dispersion (Figure 4). This trend is common for nanoparticles encapsulating hydrophobic core materials made via FNP and has been attributed to an increase in the growth rate of the particle core relative to the nucleation rate [10,26]. This result indicates that TA-chitosan interactions and subsequent precipitation occurs on the time scale of nanoparticle assembly (milliseconds). Thus, nanoparticle assembly is comparable to hydrophobic core materials. 

Next, we investigated particle size stability. We note that FNP of TA with PS-*b*-PEG initially forms ~100 nm particles comparable to previous reports [15]. In the mixed solvent, the size of both TA and TA-chitosan particle (1:1 ratio of core to block copolymer) dispersions are stable for at least 1 week. While TA initially forms particles, the particles are not stable when dialyzed to remove the organic solvent as macroscopic precipitates were observed within 24 h of storage at 4 °C or at room temperature [15]. Interestingly, when dialyzed to remove the minor amount of acetone, the TA-chitosan particles were stable at pH 2 although chitosan is soluble at this pH. We attribute TA-chitosan particle stability at pH 2 to TA physically crosslinking the chitosan via multiple weak hydrogen bonds reducing its solubility at acidic pH which has been previously described [27]. No evidence of covalent crosslinking has been observed by infrared spectroscopy [27]. Unequivocal evidence of hydrogen bonding is convoluted by the presence of tannins naturally present in chitosan [27]. Overall, our results indicate that TA-chitosan interactions facilitate nanoparticle self-assembly and stability at acidic conditions by precipitating and physically crosslinking the chitosan, respectively.

We further investigated particle stability as a function of pH. The particle size increased ~33% at pH 4 compared to pH 2. The particle swelling at pH 4 can be attributed to increased neutralization of TA; the pKa of TA is reported to be 5.12 and protonization/deprotonization can occur over a broad pH range due to the proximity of the ionizable groups. Below the pKa, the phenolic protons are available for hydrogen bonding and the TA acts as a non-covalent crosslinker for chitosan reducing its solubility in water [27]. Above the pKa, the ionized phenols electrostatically interact with the cationic chitosan. 

While the particle size was stable at pH 7.4, with further increases in pH to above 12, the particle size was unstable (Table 1). The particle size instability at pH > 12 can be attributed to complete (>99%) deprotonation of chitosan (pKa 6.5–7 [28]) i.e. loss of positive charge. At pH 12, there were not sufficient hydrogen bond interactions or electrostatic interactions to stabilize the nanoparticle core due to deprotonization of TA and chitosan. Given this pH dependent stability, this may be a valuable platform for controlled release applications.

Finally, we investigated the ability to incorporate hydrophobic materials into the TA-chitosan nanoparticle. Specifically, we included nile red, a lipophilic fluorescent dye, in the formulation. Based on previous studies encapsulating nile red via FNP [23], the dye was dissolved in the organic stream with the TA and block copolymer at a total solids concentration 40 mg/mL, mass ratio of chitosan to TA of 1:1, and mass ratio of BCP to core materials of 1:1. Incorporating the dye did not significantly affect particle size, particle size distribution or zeta potential. With storage at 4 °C, no macroscopic precipitation of nile red from the dispersion was evident. In contrast, when nile red was added to TA-chitosan nanoparticles, precipitation of the dye within 24 hours was observed. These results indicated that the dye was successfully incorporated into the TA-chitosan nanoparticles during mixing. 

Measuring the fluorescence spectra confirmed the presence of the dye (Figure 5). Interestingly, the peak fluorescence occurred at 635 nm (λ_ex_ = 530 nm). This result is similar to previous reports of nile red/PS-*b*-PEG nanoparticles; Akbulut et al. [23] observed the maximum fluorescence at 630 nm. The slight red shift can be attributed to the higher dielectric constant of chitosan compared to polystyrene [29]. Thus, this result suggests that the TA-chitosan nanoparticles provide a unique microenvironment for incorporating hydrophobic moieties. 

## 4. Conclusions

Overall, we demonstrated the ability to use Flash NanoPrecipitation to rapidly (i.e., within seconds) self-assemble and physically crosslink PEGylated chitosan nanoparticles in a single step process. Specifically, confined impinging jets were used to mix chitosan (dissolved in HCl) with TA in the presence of an amphiphilic block copolymer stabilizer. TA precipitates the chitosan during mixing to initiate nanoparticle self-assembly; the precipitate is stabilized by adsorption of the hydrophobic block of the block copolymer while the hydrophilic PEG block sterically stabilizes the nanoparticle. The size of the nanoparticles can be tuned by varying formulation parameters including the total solids concentration, and block copolymer to core mass ratio. After nanoparticle self-assembly, TA also stabilizes the resulting particle by physically crosslinking the chitosan via multiple weak hydrogen bonds reducing its solubility at acidic pH. The resulting particles are stable at pH 2; the size stability depends on the protonation of the chitosan and TA swelling at pH 4 and basic pH. Finally, we demonstrate that hydrophobic moieties can be incorporated into the nanoparticle using nile red as a model compound. Given ability to incorporate hydrophobic materials and the pH dependent particle stability, these nanoparticles may be useful for controlled release applications.

## Figures and Tables

**Figure 1 polymers-11-00749-f001:**
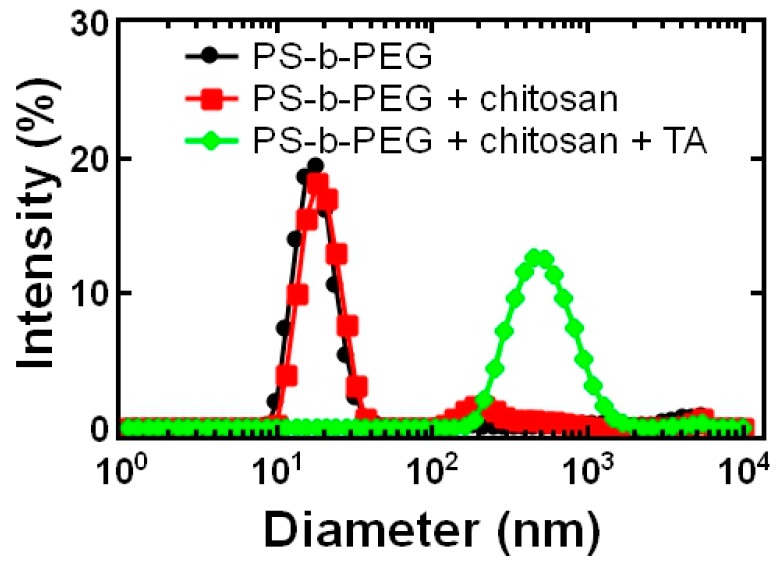
Dynamic light scattering (DLS) results of the block copolymer, polystyrene-b-polythylene glycol (PS-*b*-PEG) stabilizer that forms ~30 nm micelles (black) compared to FNP of chitosan and PS-*b*-PEG (red) which forms block copolymer micelles with soluble chitosan. Performing FNP with tannic acid (TA) to precipitate the chitosan with PS-*b*-PEG to stabilize the precipitate results in uniform particles 920 ± 150 nm with a polydispersity index (PDI) of 0.21 ± 0.02 Water was used a solvent and the nominal nanoparticle concentration of approximately 0.3 mg/mL.

**Figure 2 polymers-11-00749-f002:**
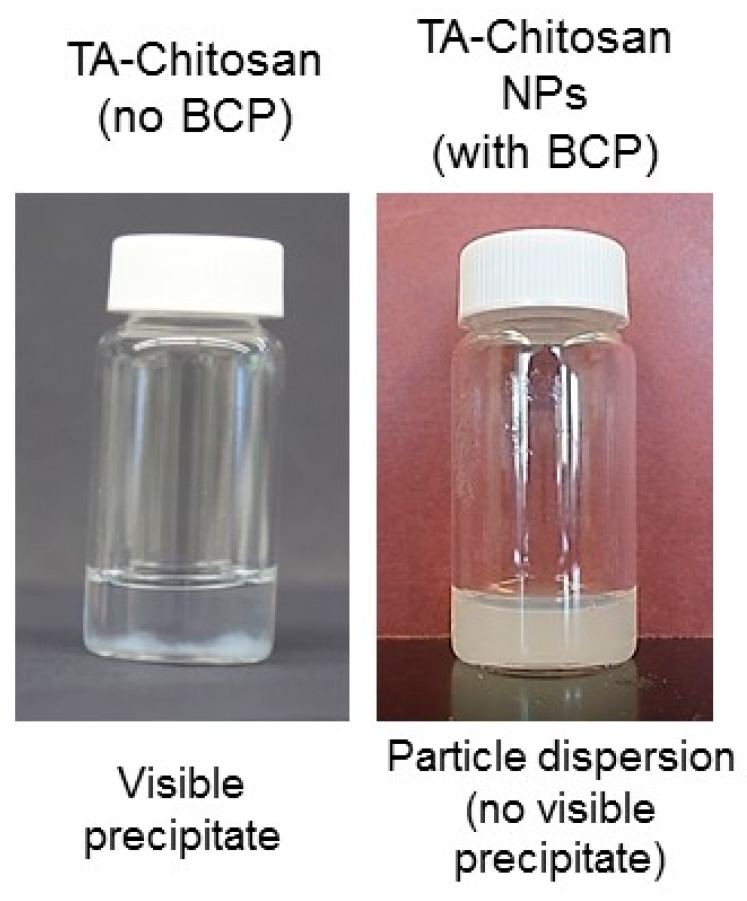
Macroscopic (visible) precipitates form upon mixing tannic acid (TA) and chitosan without including a block copolymer (BCP) stabilizer (left). In contrast, a particle dispersion with no visible precipitate was achieved in the presence of the BCP stabilizer to prevent precipitation of the TA-chitosan complex.

**Figure 3 polymers-11-00749-f003:**
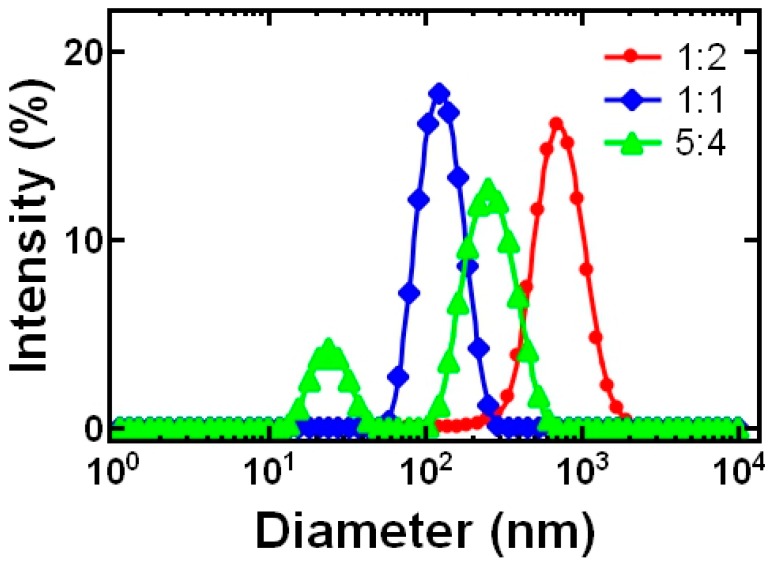
DLS results showing the effect of block copolymer:core (by mass) on tannic acid (TA)-chitosan nanoparticle size at a 1:1 ratio of TA:chitosan. Water was used as the solvent. Increasing the relative amount of block copolymer, i.e., changing the ratio from 1:2 (red) to 1:1 (blue) reduced particle size. Further increasing the relative amount of block copolymer ratio from 1:1 (blue) to 5:4 (green) resulted in the formation of TA-chitosan nanoparticles and empty micelles. The mass ratio of TA:chitosan was held constant at 1:1.

**Figure 4 polymers-11-00749-f004:**
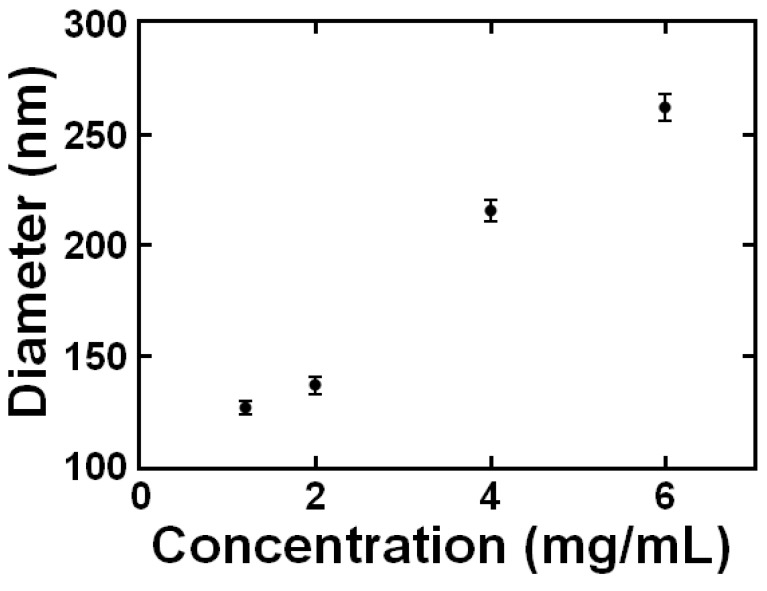
Tannic acid (TA)-chitosan nanoparticle size immediately following FNP as a function of total solids concentration (in the final dispersion). The mass ratio of TA to chitosan was 1:1; the mass ratio of block copolymer to core materials was 1:1; the solvent was water.

**Figure 5 polymers-11-00749-f005:**
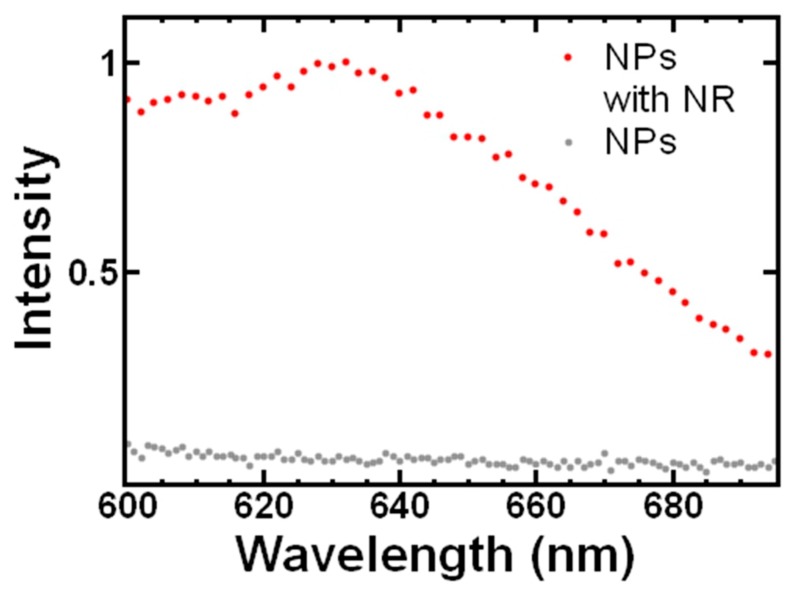
Fluorescence emission spectra (normalized) of fluorescent dye containing nanoparticles compared to tannic acid (TA)-chitosan nanoparticles.

**Table 1 polymers-11-00749-t001:** Size (diameter) and size distribution (PDI) of tannic acid (TA)-chitosan particles as a function of pH measured by dynamic light scattering. Acidic media was achieved with the appropriate concentration of hydrochloric acid; basic media was achieved with the appropriate concentration of sodium hydroxide; pH 7.4 was achieved using phosphate buffered saline (PBS).

pH	Diameter (nm)	PDI
2	146.6 ± 2.898	0.090 ± 0.018
4	194.8 ± 8.072	0.184 ± 0.012
7.4 (PBS)	160.2 ± 14.49	0.325 ± 0.036
12	N/A	>0.6

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
