# Peer review of "Single-Step Self-Assembly and Physical Crosslinking of PEGylated Chitosan Nanoparticles by Tannic Acid"

_polymers, 2019, doi:10.3390/polym11050749_

Reviewer 1 Report

While it is a subject widely studied, where they use various biomaterials modified with nanoparticles. The contribution in this proposal is presented precisely in the summary. The introduction is presented appropriately. However, it can be improved through a thorough analysis of updated literature, so that the novelty of this proposal is validated.

Figure 1. The image is not clear. The description must be precise, identifying the contribution of said results. It is presented in a descriptive way.
In line 117: This result suggests that .... a more proifundant analysis with scientific support is recommended.

In lines 131-133. This result also suggests that the nanoparticles are......Expand discussion on this.

Lines 140-141. Because in the presence of chitosan, the resulting TA-chitosan nanoparticles are larger than TA nanoparticles. Improve analysis with scientific support.

Description of figure 3 is confusing.

In general, the authors gave discussion, but the discussion was not adequate and compared with similar authors. The authors used and cited some references, but it is not enough for suitable scientific discussion. The conclusion can be improved. This is described in a general way.

Reviewer 2 Report

The manuscript is devoted to detailed description of preparation of crosslinked, self-assembled chitosan nanoparticles incorporating  hydrophobic materials using Flash Precipitation with tannic acid and the results are compared with data reported in previous publications by other authors. These results are useful for the further development of methods of nanoparticles preparation for biomedical applications using Flash NanoPrecipitation.

The word "then" repeated second time in line 50.

Author Response

We thank the reviewer for their attention to detail and corrected the repeated word.

Reviewer 3 Report

The manuscript describes self-assemble of pgylated-chitosan nanoparticles with tannic acid.  The title does not reflect the content and the materials lack detail needed to reproduced the work. What is meant by water with solvent? The nanoparticles with dye were diluted in the characterization of the nanoparticles, how and why? 

Could you please explain the interaction of the copolymer with tannic acid, and the formation of nanoparticles? What would be the relevance of using this combination?

The developed nanoparticles will be used in what application?

The DLS data should be presented with the adequate algarisms. Please clarify the solvent of the samples? were all measured in PBS?

The time required to obtained the nanoparticles should be mentioned.

Evidence of crosslinking should be shown by spectroscopic techniques.

Author Response

Round  2

Reviewer 1 Report

The article has been improved. The discussion and analysis based on the scientific literature was evidenced in the corrections made. It would be appropriate to improve the quality of some images and strengthen the conclusions, they are not clear.

Reviewer 3 Report

The authors have conducted an appropriate revision of the manuscript, and it is recommended for publication. 

Author Response

We thank the reviewer for their service and are delighted that they recommend the revised manuscript be published.